# Gas diffusion enhanced electrode with ultrathin superhydrophobic macropore structure for acidic $CO_2$ electroreduction

Mingxu Sun [1], Jiamin Cheng[2] & Miho Yamauchi [1,2,3,4,5] ✉

Carbon dioxide ($CO_2$) electroreduction reaction ($CO_2RR$) offers a promising strategy for the conversion of $CO_2$ into valuable chemicals and fuels. $CO_2RR$ in acidic electrolytes would have various advantages due to the suppression of carbonate formation. However, its reaction rate is severely limited by the slow $CO_2$ diffusion due to the absence of hydroxide that facilitates the $CO_2$ diffusion in an acidic environment. Here, we design an optimal architecture of a gas diffusion electrode (GDE) employing a copper-based ultrathin super-hydrophobic macroporous layer, in which the $CO_2$ diffusion is highly enhanced. This GDE retains its applicability even under mechanical deformation conditions. The $CO_2RR$ in acidic electrolytes exhibits a Faradaic efficiency of 87% with a partial current density ($j_{C_{2+}}$) of −1.6 A cm$^{-2}$ for multicarbon products ($C_{2+}$), and $j_{C_{2+}}$ of −0.34 A cm$^{-2}$ when applying dilute 25% $CO_2$. In a highly acidic environment, $C_{2+}$ formation occurs via a second order reaction which is controlled by both the catalyst and its hydroxide.

Achieving effective utilization of carbon dioxide ($CO_2$) is of great significance for reducing the negative impacts of climate change and mitigating crisis caused by fossil fuel shortages[1]. Renewable electricity-driven $CO_2$ reduction reaction ($CO_2RR$) to multicarbon ($C_{2+}$) products are highly demanded due to their high availability in various fields and current market price[2]. Over the past few years, a flow-type reactor has been used to investigate the $CO_2RR$ performance[3], where gas reactants diffuse directly into catalyst-electrolyte interfaces through a gas diffusion layer (GDL), where $CO_2RR$ can occur at a high current density ($j$)[4–7].

Nevertheless, most reports have used strong alkaline electrolytes to ensure high $C_{2+}$ selectivity in $CO_2RR$, which causes severe non-Faradaic consumption of $CO_2$ due to preferential carbonate formation under alkaline conditions ($CO_2 + OH^- \rightarrow CO_3^{2-}$) and results in lower carbon utilization[8,9]. Moreover, the long-term accumulation of carbonate leads to flooding of the gas diffusion electrode (GDE), which thereby terminates the reaction (Supplementary Fig. 1). The operation of $CO_2RR$ under acidic conditions therefore offers entirely different

strategy to overcome the challenges of conventional alkaline electrolyte systems[10].

In recent years, the development of highly selective catalysts that can effectively suppress the hydrogen evolution reaction (HER) in acidic environments has become a research hotspot. While some advanced catalysts have made progress in mitigating HER, they suffer from a significant reduction in the $CO_2RR$ rate compared to alkaline environments[11,12]. The primary reason for this matter is the absence of a hydroxide ion ($OH^-$) in an acidic environment, which causes insufficient adsorption of acidic $CO_2$ gas molecules and limited diffusion of $CO_2$ to the catalyst-electrolyte interface, resulting in the prevalence of undesired HER, particularly at high $j$[10–12].

To achieve high $j$, it is critical to overcome these inherent limitations in $CO_2RR$ under acidic conditions. One potential approach is to increase the $CO_2$ concentration at the catalyst-electrolyte interface by increasing the $CO_2$ flux in the gas diffusion layer (GDL), which could serve as a simple and effective strategy to improve $CO_2RR$ efficiency. Furthermore, hydroxide (OH)-derived Cu catalysts (OH-Cu) have been

[1]Department of Chemistry, Graduate School of Science, Kyushu University, Nishi-ku, Fukuoka, Japan. [2]Research Center for Negative Emissions Technologies (K-NETs), Kyushu University, Nishi-ku, Fukuoka, Japan. [3]Institute for Materials Chemistry and Engineering (IMCE), Kyushu University, Nishi-ku, Fukuoka, Japan. [4]International Institute for Carbon-Neutral Energy Research (WPI-I²CNER), Kyushu University, Nishi-ku, Fukuoka, Japan. [5]Advanced Institute for Materials Research (WPI-AIMR), Tohoku University, Aoba-ku, Sendai, Japan. ✉e-mail: yamauchi@ms.ifoc.kyushu-u.ac.jp

considered promising for achieving high selectivity towards $C_{2+}$ compounds due to the presence of OH in the catalyst structure[13,14]. The OH-rich nature of the catalysts can help to increase the pH of the catalytic layer, effectively suppressing HER in acidic environments. In addition, GDEs are generally composed of brittle and inflexible carbon materials[15], making GDEs a non-recoverable consumable.

In this study, we conducted a systematic evaluation of the factors influencing $CO_2$ diffusion in the GDL. Based on our findings, we designed an all-metal gas diffusion enhanced Cu electrode (Cu-GDL) using OH-Cu as a catalyst. This Cu-GDL demonstrated both mechanical flexibility and applicability, making it suitable for various practical applications. We systematically evaluated the $CO_2RR$ performance in an acidic environment by optimizing the electrolyte pH, catalyst amount, $CO_2$ flow rate, electrolyte type and $CO_2$ concentration. We achieved a high Faradaic efficiency of 87% ($FE_{C_{2+}}$) with a partial current density ($j_{C_{2+}}$) of −1.6 A cm$^{-2}$ for $C_{2+}$ products, and a $j_{C_{2+}}$ of −0.34 A cm$^{-2}$ that meets industrial applications even with diluted 25% $CO_2$. We discovered unique kinetics of $CO_2RR$ on Cu-GDL under acidic conditions: the first-order reaction at pH 6 and the second-order reaction at pH 1. Moreover, the CO produced in $CO_2RR$ could be further reduced to $C_{2+}$ at slow $CO_2$ flow rates, leading to an enhancement of the single-pass

conversion efficiency (SPCE) to 42%, while simultaneously increasing $FE_{C_{2+}}$ to 87%.

## Results

### Modeling for $CO_2$ diffusion in a GDL

We first verified requirements for achieving a favorable $CO_2$ diffusion efficiency ($CO_2DE$) based on the architecture of the GDL, which is a hydrophobic porous electrode. $CO_2DE$ for a GDL is described by considering Fick's second law, Knudsen self-diffusion models and surface hydrophobicity (Supplementary Figs. 2-4, details in Methods). We found that three aspects are crucial to enhance $CO_2DE$ (Fig. 1a): (1) Thin GDL; The $CO_2$ concentration ($C_{CO_2}$) is inversely proportional to the diffusion distance ($\Delta x$), and high $C_{CO_2}$ is realized at a site with small $\Delta x$ as shown in Fig. 1b. (2) Large pore diameter; Given the frequent collisions between $CO_2$ molecules and nano/micropore walls (Fig. 1c)[16], $C_{CO_2}$ is halved when the porous diameter is smaller than 128 nm, followed by a severe $C_{CO_2}$-limiting behavior with decreasing pore diameter (Fig. 1d, Supplementary Fig. 3). (3) Superhydrophobic structure; Considering that $CO_2$ diffusion coefficient in the gas phase is approximately four orders of magnitude higher than that in the liquid phase[17,18], a hydrophobic interface formed between the catalyst surface

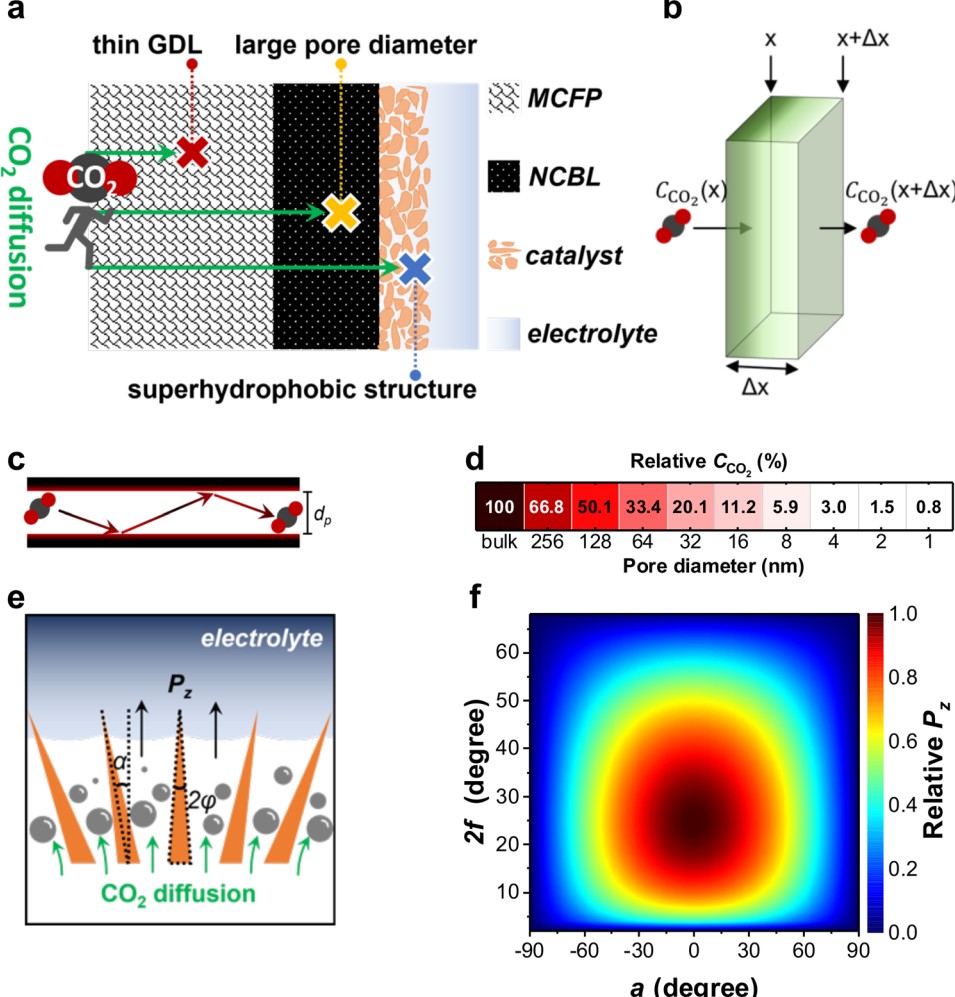

**Fig. 1 | Modeling for $CO_2$ diffusion in a GDL. a** Illustrations of $CO_2$ diffusion in the most common GDL. The thickness and pore diameter of the GDL, and the hydrophobicity of the catalyst together affect the $CO_2$ diffusion. (MCFP is macroporous carbon fiber paper and NCBL is nano-microporous carbon black layer). **b** $CO_2$ concentration ($C_{CO_2}$) decreasing along diffusion distance ($\Delta x$). **c** Knudsen model for $CO_2$ diffusion through a pore with diameter ($d_p$). **d** Relative $C_{CO_2}$ calculated toward $d_p$. **e** Illustration of $CO_2$ diffusion on a needle-like architecture of the catalyst-electrolyte interface. $P_z$ is Laplace pressure, $2\varphi$ is apex angle and $\alpha$ is tilt angle of the needle. **f**, Relative $P_z$ calculated toward $2\varphi$ and $\alpha$.

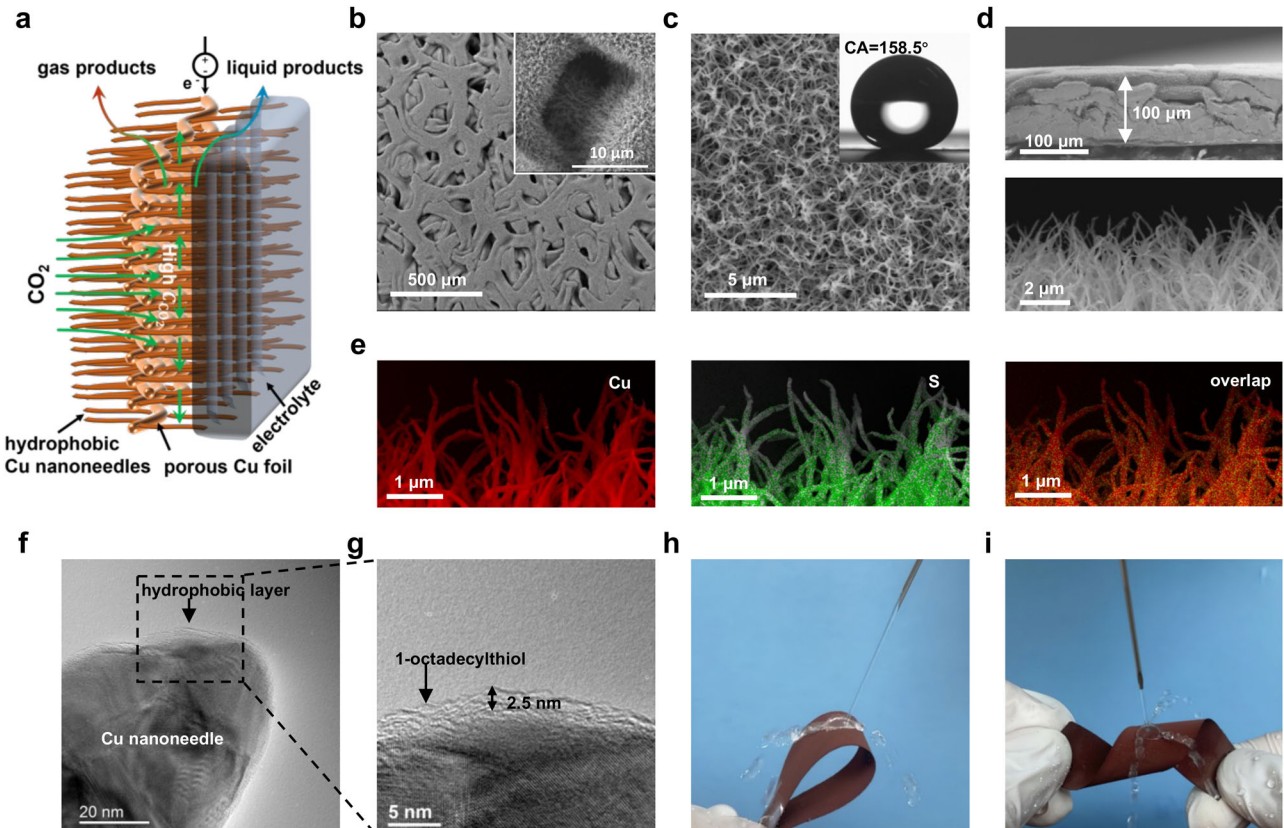

**Fig. 2 | Structural and compositional features of Cu-GDL. a** Illustration of Cu-GDL applied in $CO_2RR$. **b**, **c** SEM images for **b** macropores (the inset of **b** is the macroporous channel) and for **c** high-density nanoneedles exhibiting a large contact angle (CA). **d** Cross-sectional SEM images to confirm ultrathin electrode thickness and vertically grown nanoneedles. **e** Elemental distribution. (red is Cu, green is S). **f**, **g** TEM images covered with ~2.5 nm 1-octadecylthiol single layer as a hydrophobic layer. **h** Bending resistance test. **i**, Twisting resistance test.

and an electrolyte is an ideal space for the $CO_2$ diffusion. The positive Laplace pressure ($P_z$) of a gas phase at an interface between liquid and nano/microscale texture is a measure for emergence of hydrophobicity (Supplementary Fig. 4)[19]. Thus, $CO_2$ can rapidly diffuse on catalyst-electrolyte interfaces characterized with a positive $P_z$ (Fig. 1e). According to the Laplace equation, $P_z$ always shows a positive value when the tilt angle is $-90° < \alpha < +90°$ and the apex angle is $0° < 2\varphi < 66°$ (Fig. 1f). This assessment of GDL architectures led us to design a GDE with a thinner, macroporous diameter, as well as vertical and needle-like surface structures to facilitate $CO_2$ diffusion.

## Structural and compositional features of Cu-GDL

GDEs are commonly prepared using nanoparticulate electrocatalysts and binders on GDL substrates[20], but they do not meet the above requirements. In contrast, a porous Cu electrode, prepared by an in situ electrooxidation of a porous Cu foil, is sufficiently thin and has large pores, with its surface covered by special nanostructures. However, achieving vertical and needle-like structures with characteristic apex and tilt angles requires precise control of the incorporation rates of $Cu^{2+}$ and $OH^-$ on the Cu foil surface. By optimizing the synthetic conditions, we found that the balance between electrolyte concentration (KOH) and $j$ for the electrooxidation of the Cu foil surface is the most important factor to construct favorable nanostructures (see Methods and Supplementary Figs. 5–7 for more details). The reaction temperature (0 °C) is also critical factor for achieving the vertical and needle-like structure with characteristic apex and tilt angles (see Methods and Supplementary Figs. 8 and 9 for more details). However, the as-prepared porous Cu easily

penetrates the aqueous solution due to its large pore structure, which is unsuitable for use as a GDE in a flow cell. Therefore, we imparted water-barrier properties on the porous Cu surface by coating with 1-octadecanethiol, resulting in Cu-GDL[21].

A schematic diagram of Cu-GDL with enhanced $CO_2$ diffusion in $CO_2RR$ is shown in Fig. 2a. A scanning electron microscopy (SEM) image of Cu-GDL confirmed that the macroporous foil structure remains intact (Fig. 2b and Supplementary Fig. 10) and that nanoneedles are grown highly densely (Fig. 2b, c). A cross-sectional SEM image of Cu-GDL confirmed the ultrathin structure of ~100 μm (Fig. 2d), which is ~0.4 times thicker than a commercial carbon GDL (Supplementary Fig. 2). Energy dispersive X-ray (EDX) mapping analysis revealed that small amount (0.11 wt%) of S element of alkanethiol is uniformly distributed on the surface of Cu nanoneedle (Fig. 2e, Supplementary Fig. 11). Transmission electron microscopy (TEM) of Cu-GDL revealed that a Cu nanoneedle was coated with a layer of 1-octadecanethiol with a thickness of ~2.5 nm (Fig. 2f and g), which corresponds to the chain length of 1-octadecanethiol between the surface-bound S and the terminal C[21]. It should be noted that X-ray photoelectron spectroscopy (XPS) for Cu $2p$ revealed that the Cu nanoneedle coated with a layer of 1-octadecanethiol does not show any effect on the oxidation state of Cu (Supplementary Fig. 12). In addition, attenuated total reflectance Fourier transform infrared (ATR-FTIR) spectroscopy (Supplementary Fig. 13), XPS for S $2p$ (Supplementary Fig. 14), and microscopic observations (Supplementary Fig. 15) further confirmed the existence of 1-octadecanethiol on Cu-GDL and its roles. The perpendicularly grown needle-like nanostructures on the surface were observed by SEM (Fig. 2d). Such high-density nanoneedle structures could multiply the hydrophobicity according to the $P_z$ (see Eq.

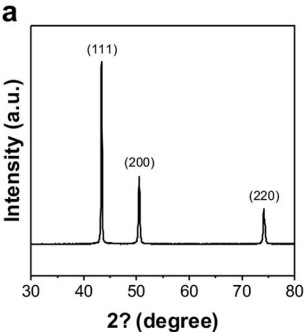
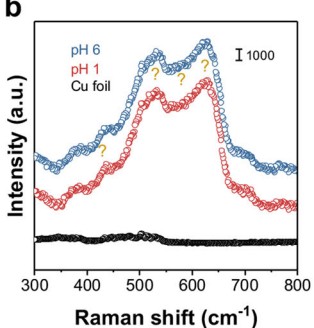
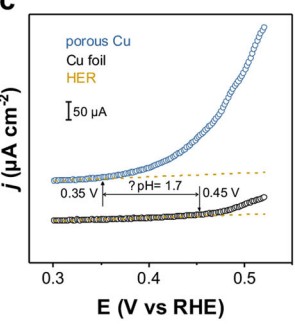

**Fig. 3 | Surface states of Cu-GDL. a** XRD pattern of Cu-GDL. **b** In situ Raman spectroscopy for Cu-GDL at pH 6 and pH 1 with E of −1.5 V vs. SHE (Cu foil used as reference baseline, yellow stars mark bands associated with OH). **c** Evaluation of surface pH on Cu in 0.1 M Ar-saturated KHCO₃. A yellow dashed curve represents a fitted line for the HER current. The onset oxidation potential of a Cu foil is 0.45 V (vs. RHE), and porous Cu is 0.35 V (vs. RHE). The pH difference (ΔpH) between a Cu foil and porous Cu is 1.7.

(4)). The apex angle was determined to be in the range of 18–25° (Supplementary Fig. 16), which is very close to the perfect value of 22° suggested by our calculation of relative $P_z$ versus apex angles (Fig. 1f). A contact angle (CA) measurement showed that Cu-GDL has super-hydrophobicity characterized with a CA of 158.5° (Fig. 2c). In contrast, an alkanethiol-modified original Cu foil exhibited normal hydrophobicity with a CA of 123.3°, which clearly indicates that super-hydrophobicity of Cu-GDL originates from its surface structure and not only from the coating with the thiol layer (Supplementary Fig. 17). In addition, the nanostructures appear to be able to minimize the size of bubbles to enhance their desorption[22,23], which is critical in terms of energy efficiency. Slow bubble removal from the electrodes is known to cause energy losses of up to 20%[24], and therefore bubbles must be efficiently removed from GDL. It should be noted that our Cu-GDL retained its superhydrophobicity even after being severely bent (Fig. 2h) and twisted (Fig. 2i). Such a durable and flexible super-hydrophobic GDL may contributes to the industrialization of CO₂ electroreduction.

## Surface states of Cu-GDL

X-ray diffraction (XRD) patterns confirmed that the major component of Cu-GDL is metallic Cu (Fig. 3a). Interestingly, ex situ Raman spectroscopy revealed similar Raman signals around 431, 528, and 623 cm⁻¹ for both Cu-GDL and Cu(OH)₂ (Supplementary Fig. 18)[25]. However, no peaks related to Cu²⁺ were observed in the XPS for Cu 2p (Supplementary Fig. 12). Furthermore, XPS for O 1s revealed the presence of OH on the surface of Cu-GDL (Supplementary Fig. 19)[14]. These findings indicate that the composition of Cu-GDL consists mainly of metallic Cu covered with OH. To investigate the effect of acidic electrolyte on the electrode composition in CO₂RR, in situ Raman spectroscopy was conducted at a potential (E) of −1.5 V vs SHE using electrolytes of pH 6 and pH 1, which suggested that OH is stably present on the Cu-GDL even in a highly acidic solution under the potential (Fig. 3b). Given various influential factors such as current density, diffusion layer thickness, bulk electrolyte composition, and other factors, the determination of a pH just above the electrode surface, "intrinsic pH", during CO₂RR appears to be challenging[26]. To address this, we applied an electrochemical approach based on the onset potential for the oxidation of Cu⁰ to Cu¹, and the intrinsic pH on Cu-GDL was calculated from the onset potential observed on a pristine Cu foil as reference using Nernst equation (see Methods and Supplementary Fig. 20 for details). In this experiment, we used a slow scan rate of 10 mV s⁻¹ at low j in the range of μA cm⁻², in an inert and near-neutral 0.1 M Ar-saturated KHCO₃ electrolyte. The test revealed that the catalyst surface has a pH of 10.9 (Fig. 3c), suggesting the alkaline nature of the Cu-GDL surface, which would reduce local H⁺ concentrations, and thereby suppress HER during CO₂RR.

## CO₂RR performance of Cu-GDL and underlying mechanisms

The CO₂RR performance was evaluated at *j* ranging from −0.3 to −1.8 A cm⁻² by using acidic electrolytes. High $FE_{C_{2+}}$ of 87% and low Faradaic efficiency for HER less than 4% were achieved at both pH 6 and pH 1 (Fig. 4a, Supplementary Fig. 21). The Cu-GDL retains its original morphology after CO₂RR operation (Supplementary Fig. 22).

We first optimized the catalyst amount by changing the reaction time for the electrooxidation of a Cu foil. The original Cu foil coated with 1-octadecanethiol hardly participated in CO₂RR, and both the $j_{C_{2+}}$ and the formation rate of $C_{2+}$ ($FR_{C_{2+}}$) tended to increase with increasing the electrooxidation time (Supplementary Figs. 23 and 24). To understand the relationship between the catalytically active surface area and $FR_{C_{2+}}$, we further evaluated the roughness factors (rf) of each electrode by analyzing electrochemical surface area (ECSA), which reflects the number of catalytic sites (Supplementary Fig. 25). The SEM images of the electrodes corresponding to the rf are shown in Fig. 4b. Considering that the high selectivity of Cu-GDL towards $C_{2+}$ is accompanied by minimal HER, we established the relationship between the $FR_{C_{2+}}$ and rf (Supplementary Fig. 26). Interestingly, there was a good linear correlation between $FR_{C_{2+}}$ and rf at pH 6, whereas $FR_{C_{2+}}$ showed nonlinear behavior at pH 1 (Fig. 4b, Supplementary Fig. 26). Since the as-prepared Cu-GDL has an OH-rich surface, which reduces the local H⁺ concentration, we reasoned that not only the active surface area but also the amount of OH present on the catalyst simultaneously determines the $FR_{C_{2+}}$ at pH 1 as described in Fig. 4c. We then attempted tried to analyze the relationship between $FR_{C_{2+}}$ and the quadratic roughness factor (rf²). Surprisingly, $FR_{C_{2+}}$ showed a very good linear correlation with rf² at pH 1 (Fig. 4b). Although the kinetics of CO₂RR may involve various influencing factors, we made a predictive investigation of CO₂RR kinetics in an acidic environment by precisely controlling the variables and found that CO₂RR on Cu-GDL at pH 6 is a first-order reaction depending on the number of catalytic sites, whereas CO₂RR at pH 1 can be described as a second-order reaction which is determined by the number of catalytic sites and the amount of OH. In this regard, we further characterized OH concentration on each electrode (Supplementary Fig. 27) and determined the rate constants of CO₂RR (*k*) at pH 6 and pH 1 to be 4.20 × 10² h⁻¹ and 2.45 × 10¹ μmol⁻¹ cm² h⁻¹, respectively (Supplementary Fig. 28)[27].

Next, we evaluated the carbon utilization in CO₂RR at −0.5 A cm⁻² with CO₂ flow rate ranging from 10 to 3 sccm. Considering that the possibility to improve the recovery of CO₃²⁻ to CO₂ during CO₂RR, which has been considered as a key to enhance CO₂ utilization[12], we additionally employed two sets of electrolytes; CO₂-saturated electrolytes with pH 6 and pH 1, namely pH 6_CO₂ and pH 1_CO₂. The carbon utilization tests suggested that both SPCE and conversion efficiency for $C_{2+}$ ($CE_{C_{2+}}$) gradually increase as the CO₂ flow rate decreases (Fig. 4d, Supplementary Fig. 29), and the corresponding

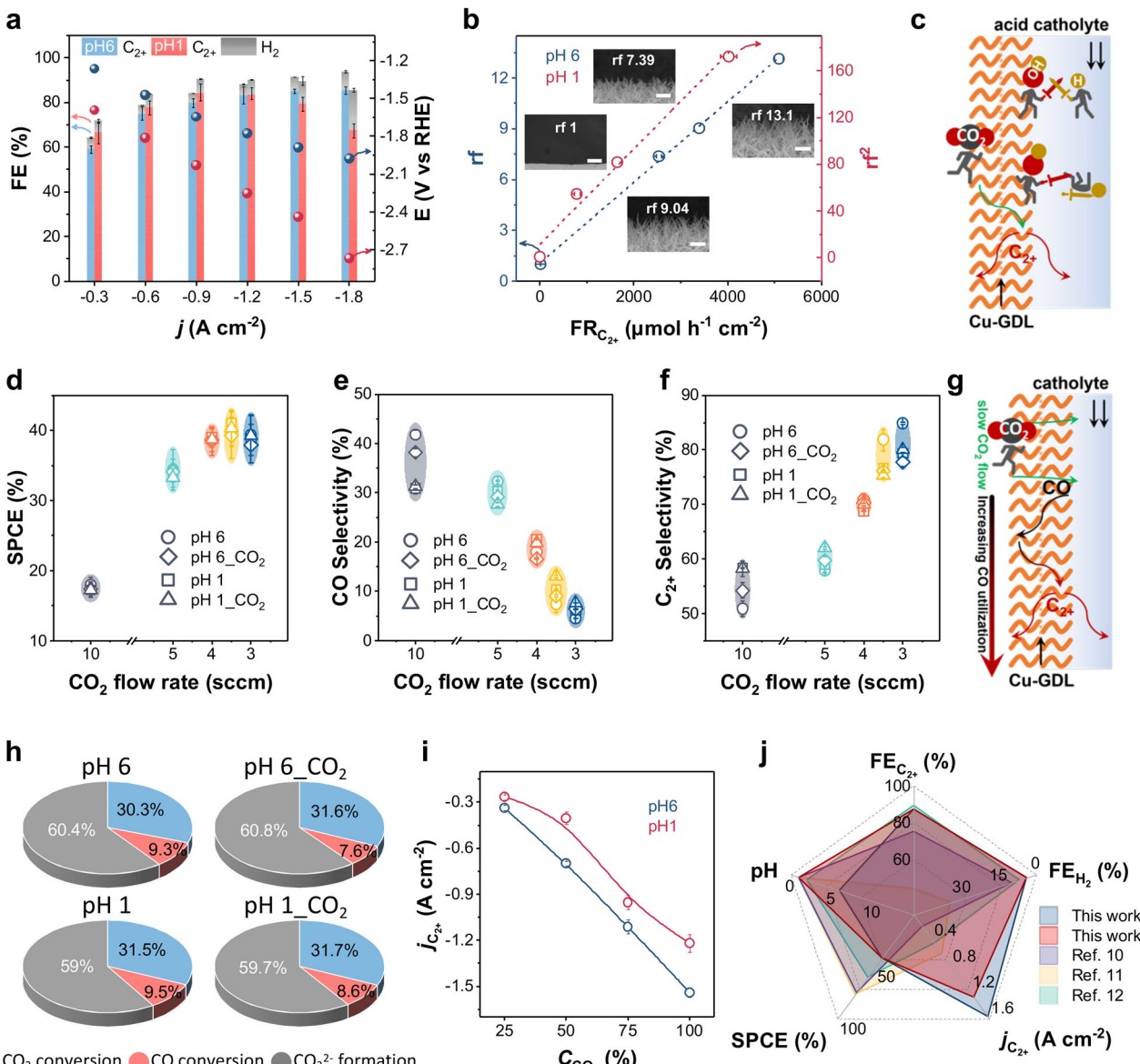

**Fig. 4 | CO₂RR performance of Cu-GDL and underlying mechanisms. a** FE for the production of $C_{2+}$ and $H_2$, and potential (E vs. RHE, right) as functions of $j$ (pH 6 is blue, pH 1 is red). **b** rf (left) and $rf^2$ (right) as functions of $FR_{C_{2+}}$. Inset shows the SEM images of Cu-GDL having corresponding rf (error bars = 2 μm). **c** Illustration of a function of OH species on Cu-GDL during CO₂RR in acidic media. **d** SPCE, **e** CO selectivity and **f** $C_{2+}$ selectivity as a function of CO₂ flow rate at an applied $j$ of −0.5 A cm⁻² using electrolytes with pH = 6 (pH 6) and pH = 1 (pH 1) and CO₂

saturated electrolytes with pH = 3.8 (pH 6_CO₂) and pH = 0.9 (pH 1_CO₂). **g** Illustration of secondary CO conversion during CO₂RR at a slow CO₂ flow rate. **h** Carbon utilization percentage at CO₂ flow rate of 3.5 sccm and an applied $j$ of −0.5 A cm⁻² at varying electrolyte. **i** $j_{C_{2+}}$ as a function of $C_{CO_2}$ (diluted gas is N₂). **j** Comparison of $FE_{C_{2+}}$, $FE_{H_2}$, $j_{C_{2+}}$, SPCE and pH of bulk electrolyte on Cu-GDL with those on the state-of-the-art CO₂RR in acidic media. **a, b, d, e, f, i**, error bars indicate s.d. (n = 3 replicates).

$FE_{C_{2+}}$ are shown in Supplementary Fig. 30 and 31. At a CO₂ flow rate of 3.5 sccm, it simultaneously exhibited the optimal SPCE of 42% and the optimal $FE_{C_{2+}}$ of 87% (Fig. 4d, Supplementary Fig. 31). However, when the flow rate was reduced to 3 sccm, the SPCE did not increase further beyond 42% as the CO₂ flow rate decreased (Fig. 4d). On the contrary, the HER increased to 19–26% (Supplementary Fig. 32). This result suggests that CO₂ supply via recovery from the $CO_3^{2-}$ dissolving electrolyte is not effective for improving SPCE, even when we use the $CO_3^{2-}$-rich and CO₂ saturated electrolytes. We then propose that this phenomenon can be attributed to the fact that CO₂ recovered from $CO_3^{2-}$ tends to remain in the liquid phase. The CO₂ diffusion efficiency in the liquid phase is greatly reduced compared to the gaseous phase[17,18]. The encouraging point is that, when considering the formation of $CO_3^{2-}$ during CO₂RR, our SPCE results remain above 40% for both pH 6 and pH 1 conditions, exceeding the theoretically maximum

achievable SPCE. The calculation results are summarized in Supplementary Tables 1 and 2 and used for comparison (see Methods). Further analysis of the product selectivity showed that the selectivity of CO decreases with decreasing of CO₂ flow rate (Fig. 4e), resulting in a continuous increase of the selectivity of $C_{2+}$ (Fig. 4f). This result leads us to conclude that a part of the theoretical maximum SPCE originates from the CO secondary conversion of as described in Fig. 4g. CO secondary conversion refers to the fact that the low CO₂ flow rate results in a prolonged residence time of CO, derived from CO₂ conversion, on the catalyst surface. Consequently, CO is further reduced, leading to the $C_{2+}$ formation. Importantly, an enhancement in SPCE is observed due to the absence of side reactions involving $CO_3^{2-}$ formation during the carbon monoxide reduction (CORR) process[28]. Calculated carbon utilizations in CO₂RR at a CO₂ flow rate of 3.5 sccm are summarized in Supplementary Table 2, which shows that 30.3–31.7% of

the carbon products come from $CO_2$ conversion, 7.6–9.5% from CO conversion, and 59.0–60.8% of the $CO_2$ is used for $CO_3^{2-}$ formation (Fig. 4h).

In view of the enhanced $CO_2$ diffusion properties of Cu-GDL, we further evaluated the $CO_2$RR performance in the presence of $N_2$ impurity gas (Supplementary Fig. 33). The $CO_2$RR performance indicated that the influence of $C_{CO_2}$ on the $FE_{C_{2+}}$ is limited, but remarkable on the $j_{C_{2+}}$ (Fig. 4i, Supplementary Fig. 34). The $j_{C_{2+}}$ for diluted 25% $CO_2$ gas reached −430 and −280 mA cm$^{-2}$ at pH 6 and pH 1, respectively (Fig. 4i). Even at such low $C_{CO_2}$, the productivity levels are still in a regime of interest with respect to industrial application[29]. The $CO_2$RR performance of Cu-GDL was compared with those of state-of-art $CO_2$RR in acidic media as shown in Fig. 4j[10–12]. In addition, in previous systems, the continuous accumulation of $CO_3^{2-}$ during $CO_2$RR required procedures such as stopping the reaction and replacing the electrolyte to maintain $CO_2$RR activity[4,30,31]. However, these procedures significantly increased the operating costs associated with $CO_2$RR. To address this issue, we designed an automatic electrolyte recovery system (Supplementary Fig. 35) that can maintain a $CO_3^{2-}$ free electrolyte during the $CO_2$RR operation by introducing a small amount of HCl to decompose carbonates: $CO_3^{2-} + H^+ \rightarrow CO_2 \uparrow$. By implementing the electrolyte renewal system, we were able to achieve continuous operation of $CO_2$RR at a pH ≈ 1 for more than 10 h with an applied $j$ of −600 mA cm$^{-2}$ (Supplementary Fig. 36), and for more than 30 h with an applied $j$ of −200 mA cm$^{-2}$ (Supplementary Fig. 37). In the future, the development of a more stable hydrophobic layer that surpasses the stability of thiols holds promise for further improving the long-term stability of $CO_2$RR at high $j$.

## Discussion

We first validated $CO_2$DE of the conventional GDL and found that $CO_2$DE can be improved by optimizing thickness, pore size, and hydrophobicity of GDL. Based on the analysis results, we designed a novel Cu-GDL exhibiting ultrathin, macroporous and superhydrophobic properties. It should be noted that Cu-GDL maintained its applicability even under severe bent and twisted states. By employing Cu-GDL, we achieved a $FE_{C_{2+}}$ of 87% at $j_{C_{2+}}$ of −1.6 A cm$^{-2}$ under acidic conditions and discovered that $CO_2$RR on Cu-GDL is characterized as a first-order reaction at pH 6 and a second-order reaction at pH 1. We proposed the secondary CO conversion mechanism for the enhancement of SPCE in $CO_2$RR. Furthermore, we achieved a $j_{C_{2+}}$ of −0.34 A cm$^{-2}$ even using diluted 25% $CO_2$. Overall, understanding $CO_2$RR systems on the enhanced gas diffusion electrode implemented in this work would accelerate the development of $CO_2$RR technology. In the future, it would be useful to further investigate the influence of the local environment in the gas diffusion electrode using advanced in-situ detection methods.

## Methods
### Modeling of $CO_2$ diffusion
The $CO_2$ diffusion was assessed by assuming the architecture of GDE, which is hydrophobic porous electrodes[32]. Generally, the gas concentration change due to diffusion is predicted by Fick's second law, which is determined by the following equation:

$$\frac{\partial C_{CO_2}}{\partial t} = D \frac{\partial^2 C_{CO_2}}{\partial x^2} \tag{1}$$

where $C_{CO_2}$ is the concentration of $CO_2$, $t$ is time, $D$ is the diffusion coefficient, $x$ is crosswise position of the GDE. When $C_{CO_2}$ is in a steady state, i.e. $C_{CO_2}$ does not change with $t$,

$$\frac{\partial C_{CO_2}}{\partial t} = 0 \tag{2}$$

Because a macropore structure does not limit $CO_2$ diffusion, $C_{CO_2}$ in the macroprous carbon fiber paper (MCFP, Supplementary Fig. 2) having a constant $D$ shows a linear dependence on $x$ as shown in Fig. 1b and, therefore, the Fick's second law suggests that $CO_2$ diffusion efficiency can be improved by thinning GDE.

A nano-microporous carbon black layer (NCBL) is made up of nanoscale carbon black particles that forms a nano-microporous layer (Supplementary Fig. 2), and the pore sizes are further reduced when catalyst coating is applied[5]. In light of frequent collisions between $CO_2$ molecules and nano-micropore walls, as illustrated in Fig. 1c. The diffusion coefficient ($D$) in the nano-micropore domain is determined by Knudsen diffusivity[16], which is described as follows:

$$D = \left( \frac{1}{D_b} + \frac{3}{\sqrt{\frac{8RT}{\pi M}} d_p} \right)^{-1} \tag{3}$$

where $D_b$ is the bulk diffusion coefficient of $CO_2$ ($1.6 \times 10^{-5}$ m$^2$s$^{-1}$)[5], $R$ is the gas constant, $T$ is temperature, $M$ is molecular mass of $CO_2$, $d_p$ is an average diameter of the nano-micropore. We calculated $D$ based on Eq. (3) and suggested that $C_{CO_2}$ is halved when $d_p$ becomes smaller than 128 nm, which probably causes severe $CO_2$ concentration-limiting behavior (Fig. 1d, Supplementary Fig. 3).

As $D$ values are approximately four orders of magnitude higher than those in the liquid phase[17,18], preparing a superhydrophobic catalyst-electrolyte interface can prevent decay of the $CO_2$ diffusion rate. Nature often exhibits a superhydrophobic surface formed on its nano-microtextured structure[33], such as lotus[34], rice leaves[35], water striders' legs[36] and moth eyes[37]. A superhydrophobic surface shows a positive Laplace pressure ($P_z$) from the gas present in a nano-microscale texture[19]. The $P_z$ can be expressed by using the following equation[30]:

$$P_z = 2\gamma \sqrt{\pi \Omega f \sin\varphi} \left[ \sin\left(\theta_0 - \frac{\pi}{2} - \varphi + \alpha\right) + \sin\left(\theta_0 - \frac{\pi}{2} - \varphi - \alpha\right) \right] / 2$$
$$= \gamma \sqrt{\pi \Omega f \sin\varphi} \cos\alpha \sin\left(\theta_0 - \frac{\pi}{2} - \varphi\right) \tag{4}$$

where $\gamma$ is the surface tension of the liquid, $\Omega$ is the density of the nano-microscale texture, $f$ is the adhesion fraction of the liquid–solid interface, $\varphi$ is half-angle of the apex, $\alpha$ is the tilt angle of the needle (Fig. 1e, Supplementary Fig. 4), and $\theta_O$ is the static contact angle (123.3°), which obtained on surface of a porous Cu foil modified with 1-octadecanethiol (Supplementary Fig. 17). When the tilt angle is −90° < $\alpha$ < +90° and the apex angle is 0° < $2\varphi$ < 66°, $P_z$ always shows a positive value (Fig. 1f). Our calculations indicate the maximum $P_z$ value can be achieved at $\alpha$ = 0° and $2\varphi$ = 22° (Fig. 1f), which indicates that the vertical and needle-like structure reveals the marked hydrophobicity on its surface.

### Electrode preparation and optimization
The preparation of GDE is based on the optimization of a previously reported in situ electrooxidation method[14]. A porous Cu foil was first washed with HCl (36%, Wako) for 1 min, followed immediately by ultrasonic cleaning with acetone (99.5%, Wako), ethanol (99.5%, Wako), and deionized water (18.2 MΩ cm$^{-1}$) for 5 min to remove surface impurities.

Based on the principle of in situ electrooxidation, the anode reaction can be described as follows:

$$Cu \rightarrow Cu^{2+} + e^-$$
$$OH^- + Cu^{2+} \rightarrow Cu(OH)_2 \tag{5}$$

Upon examining the reaction formula, it becomes evident that the relative concentrations of $OH^-$ and $Cu^{2+}$ play a crucial role in facilitating the efficient occurrence of the electrooxidation process. Consequently, we conducted several optimizations to ensure favorable conditions. These included adjusting the KOH concentration in the electrolyte (which controls the $OH^-$ concentration, as shown in Supplementary Figs. 5 and 6), the electrooxidation current density (which controls the $Cu^{2+}$ concentration, as shown in Supplementary Fig. 7), and the electrooxidation temperature (which influences the rate of $Cu^{2+}$ and $OH^-$ combination, as shown in Supplementary Fig. 8).

After careful consideration, we decided to use a 2 M aqueous KOH (85%, Wako) electrolyte was used in an ice bath maintained at 0 °C. This setup allowed us to electrochemically oxidize the surface of the cleaned porous Cu foil, leading to the fabrication of needle-like nanostructures with characteristic apex and tilt angles. The electrooxidation process involved maintaining a constant oxidation current density of $-4\,mA\,cm^{-2}$ for approximately 30 minutes to ensure a high density of needle-like nanostructures. The resulting porous Cu foil was then dried under flowing $N_2$ for 30 minutes to form porous $Cu(OH)_2$.

To preserve the original needle-like nanostructure morphology during the reduction of the porous $Cu(OH)_2$, we implemented a mild temperature program (3 °C per minute) by gradually heating the porous $Cu(OH)_2$ to 180 °C over the course of 1 h in an $H_2$ environment. This step was critical because electrochemical reduction of the porous $Cu(OH)_2$ would otherwise destroy the original needle-like nanostructure (as illustrated in Supplementary Fig. 9). After reaching room temperature, the porous Cu was immersed in an Ar-saturated ethanol solution containing 1-octadecanethiol (10 mM, Wako) at 60 °C for 10 min. This immersion process resulted in the formation of Cu-GDL. Continuous air flushing with Ar was maintained throughout the immersion to ensure the desired result.

## Characterization
X-ray diffraction (XRD) pattern was recorded using a BRUKER D2-Phaser diffractometer at 30 kV and 10 mA using Cu-K$\alpha$ radiation ($\lambda = 1.54184$ Å). Scanning electron microscopy (SEM) images were obtained with a JEOL JSM-7900F microscope. Contact angle measurement was conducted on DM-301 machine. Transmission electron microscope (TEM) images were obtained using a JEOL JEM-ARM200CF microscope operating at 200 kV. X-ray photoemission spectroscopy (XPS) data were collected using an Al-K$\alpha$ radiation source (1486.6 eV) with a PHI 5000 Versa Probe. Attenuation total reflection-Fourier transform infrared (ATR-FTIR) spectra was collected with a Nicolet iS50 spectroscopy. Raman measurement was conducted on an inVia Raman microscope with a 785 nm and 200 µW laser and a 50× objective.

## Evaluation of electrochemical surface area (ECSA)
A 0.1 M $KHCO_3$ electrolyte saturated with Ar was used for evaluation of electrochemical surface area (ECSA) to determine the electrochemical double-layer capacitance ($C_{dl}$). An electroreduction in potential at $-0.6$ V versus RHE for 2 min was performed before ECSA evaluation. Cyclic voltammograms were measured at scan rates of 10, 20, 30, 40, and $50\,mV\cdot s^{-1}$ in the potential range of $-0.7$ to $-0.6$ V versus Ag/AgCl. Ar was purged during the measurement. A data set of anodic and cathodic current densities at $-0.65$ V versus Ag/AgCl was recorded, and the resulting capacitive currents divided by 2 were plotted versus the scan rate. The slope obtained by the linear regression analysis gave the $C_{dl}$ value.

## Evaluation of surface pH on the electrode
All Cu electrodes were reduced at $-0.6$ V versus RHE for 2 min before cyclic voltammetry (CV) was performed. The CV was conducted in an Ar-saturated 0.1 M $KHCO_3$ electrolyte at a sweep rate of $10\,mV\cdot s^{-1}$ while air in the cell was purged with Ar. The surface pH on the electrode was calculated by using following equations:

$$E(vs.RHE) = E(vs.Ag/AgCl) + 0.209V + 0.0592 \times pH_{surface} \quad (6)$$

$$pH_{surface} = pH_{bulk} + \Delta pH \quad (7)$$

where, $pH_{surface}$ is surface pH of the catalyst/electrode, $pH_{bulk}$ is pH of an electrolyte (details in the description of Supplementary Fig. 20).

## In situ Raman spectroscopy measurement
A custom-made electrochemical Raman flow cell was used for in situ Raman measurement. A thin electrolyte layer of 5 mm thickness allows the laser to directly irradiate the catalyst surface. A coiled Pt wire ($100 \times 0.5\,mm^2$) as a counter electrode, and Ag/AgCl (3.0 M NaCl) as a reference electrode were used. Nafion-117 was placed to separate the catholyte and anolyte chambers. The anolyte was 1 M KOH at flow rate of $2.5\,mL\,min^{-1}$, the catholyte was 1 M KCl with pH 6 and pH 1 at a flow rate of $2.5\,mL\,min^{-1}$, and $CO_2$ was continuously flowed into the gas chamber at a flow rate of $50\,mL\,min^{-1}$ during the test.

## Electrochemical measurements
All electrochemical measurements were performed in a gas diffusion flow reactor with a Parstat MC potentiostat (Princeton Applied Research). We used a prepared Cu-GDL as a cathode, a Nafion 117 membrane as a membrane and a Ni foam as an anode, which were positioned and clamped together via gaskets. Unless otherwise stated, a mixed solution of 1 M KCl (99.5%, Wako) and 1 M HCl, and 1 M KOH were used as a catholyte, and circulated through the electrochemical cell using a peristaltic pump with a flow rate of $2.5\,ml\,min^{-1}$, while the $CO_2$ (99.99%) flow rate was controlled to be $50\,ml\,min^{-1}$. Considering the need for a sustained and stable oxygen evolution reaction to balance the high reaction rate of $CO_2RR$, a 1 M KOH was used as the anodic electrolyte. This choice is made due to the use of Nafion 117, a cation exchange membrane, that eliminates $OH^-$ crossover to the cathodic side, which affects the pH of the cathodic electrolyte. By using the following equation, the potential versus Ag/AgCl is calibrated in relation to the reversible hydrogen electrode (RHE): $E$ (vs. RHE) = $E$ (vs. Ag/AgCl) + 0.209 V + 0.0592 × pH. The 80% iR drop was compensated by electrochemical impedance spectroscopy (EIS) under open circuit potentials. This EIS test was conducted immediately after each set of $CO_2RR$ experiments to ensure the accuracy of the internal resistance. Gas chromatography (GC, Agilent 490) was used to analyze effluent gas products extracted from the cathodic compartment, and liquid products were analyzed using high-performance liquid chromatography (HPLC, Shimadzu LC-20AD).

## Calculations for reaction kinetics
The first-order and the second-order reactions are described by following equations;

$$FR_{C_{2+}} = k_1\,[Cat.] \quad (8)$$

and

$$FR_{C_{2+}} = k_2\,[Cat.]\,[OH] \quad (9)$$

where $FR_{C_{2+}}$ is formation rate of $C_{2+}$, $k_1$ and $k_2$ are the reaction rate constant, [Cat.] and [OH] are the concentration of the catalyst and the OH contained in the catalyst, respectively.

## Calculations for Faradaic efficiency (FE) and partial current density ($j$)

The Faradaic efficiency of gas products ($FE_{gas}$) and liquid products ($FE_{liquid}$) were calculated based on the following equations:

$$FE_{gas} = \frac{nFC_i \nu P}{QRT} \times 100\% \quad (10)$$

$$FE_{liquid} = \frac{nFC_i V}{Q} \times 100\% \quad (11)$$

where $n$ is the number of transferred electrons to produce one molecule of the focusing product $i$, $F$ is the Faraday constant, $C_i$ is the concentration of the product $i$ determined by GC or HPLC, $\nu$ is the flow rate of $CO_2$, $P$, $T$ and $Q$ are pressure, temperature, and total charge followed in the experiment, $R$ is the gas constant, $V$ is the volume of the electrolyte.

The partial current density ($j_i$) was calculated using:

$$j_i = \frac{I FE_i}{S} \quad (12)$$

where $I$, $FE_i$ and $S$ are reaction current, the Faradaic efficiency of the target product $i$ and surface area of the working electrode.

## Calculation for formation rate (FR), selectivity, single pass conversion efficiency (SPCE) and $C_{2+}$ conversion efficiency ($CE_{C_{2+}}$)

The formation rate, $C_{2+}$ selectivity, single pass conversion efficiency and $C_{2+}$ conversion efficiency was calculated using:

$$FR_i = \frac{QFE_i}{nFtS} \quad (13)$$

$$Selectivity_i = \frac{\sum C_i(NFR_i)}{\sum C_{all}(NFR_i)} \times 100\% \quad (14)$$

$$SPCE = \frac{\sum C_{all}(NFR_i)}{\frac{\nu}{S}} \times 100\% \quad (15)$$

$$CE_{C_{2+}} = \frac{\sum C_{2+}(NFR_i)}{\frac{\nu}{S}} \times 100\% \quad (16)$$

$$SPCE_i = Selectivity_i \times SPCE_t \times 100\% \quad (17)$$

where $FR_i$, $Selectivity_i$, and $SPCE_i$ are formation rate, selectivity, single pass conversion efficiency of the target product $i$, respectively. $t$ and $N$ are reaction time and the number of carbon atoms in each product $i$, respectively. $C_{all}$ is sum of $C_i$. $SPCE_t$ is theoretical maximum SPCE of the target product $i$ when considering the carbonate formation.

## Data availability

Source data are provided with this paper. Source data for $CO_2$ diffusion modeling, surface state characterization, and $CO_2$ reduction reaction performance are available from Figshare with the accession code https://doi.org/10.6084/m9.figshare.24778809. Source data are provided with this paper.

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

## Acknowledgements
We acknowledge funding from Japan Science and Technology Agency grant JPMJFS2132 (M.S.), Japan Society for the Promotion of Science Grants-in-Aid for Scientific Research JP18H05517, JP22K19088 and JP23H00313 (M.Y.), and Moonshot Research and Development Program JPNP18016 (M.Y.). We thank Prof. N. Tanaka and Prof. T. Fujigaya, Department of Applied Chemistry, Kyushu University for their help with contact angle measurements. We thank Dr. T. G. Noguchi for his help with contact angle measurements. We thank Dr. A. Anzai and Dr. M. Liu for their help with TEM measurements.

## Author contributions
Conceptualization: M.S. Methodology: M.S. and J.C. Investigation: M.S. and J.C. Visualization: M.S. and J.C. Funding acquisition: M.S. and M.Y. Project administration: M.Y. Supervision: M.Y. Writing – original draft: M.S. and J.C. Writing – review & editing: M.S. and M.Y.

## Competing interests
The authors declare no competing interests.
