## [Peer Review File · Nature Communications]

Gas Diffusion Enhanced Electrode with Ultrathin Superhydrophobic Macropore Structure for Acidic CO₂ ElectroreductionReviewers' Comments:

Reviewer #1:

Remarks to the Author:

In this manuscript, the authors first validated CO₂ diffusion efficiency (CO₂DE) of the conventional GDL and found that CO₂DE can be improved by optimizing thickness, pore size and hydrophobicity of GDL. Besides, they further systematically evaluated the CO₂RR performance of the optimal GDL in an acidic environment, which exhibits a Faradaic efficiency of 87% with a partial current density (j_{C2+}) of -1.6 A cm^{-2} for multicarbon products (C₂₊). Despite having appreciable and interesting results, there are still some issues that would require a thoughtful and major revision of the manuscript:

1. It is mentioned in the manuscript that Cu-GDL was obtained by a mild temperature program (3 °C per minute) by gradually heating the porous Cu(OH)₂ to 180 °C over the course of 1 h in an H₂ environment. And then porous Cu was immersed in an Ar-saturated ethanol solution containing 1-octadecanethiol (10 mM, Wako) at 60 °C for 10 min. The results of Raman spectroscopy and XPS show that the surface of Cu-GDL is copper, and there is still a lot of -OH on the surface. Among them, hydrogen heat treatment can reduce Cu(OH)₂ to metallic copper, but how OH is produced and stable. Please provide an explanation in this regard.

2. It is worth mentioning that previous studies have found that alkali metal cations (Na⁺, K⁺, Cs⁺) play an important role in inhibiting hydrogen evolution reaction (HER) and improving the selectivity of CO₂RR to C₂₊ products in acidic electrolytes (Science 2021, 372, 1074; Nat. Commun. 2022, 13, 7596; Nat. Catal. 2022, 5, 268). However, the influence of alkali metal cations (K⁺) on the HER and CO₂RR is not mentioned in this manuscript. Please specify whether alkali metal cations contribute to CO₂RR of performance in acidic environments in this work?

3. A more detailed distribution of C₂₊ products should be provided in Fig. 4a, which is conducive to further study of whether the optimized Cu-GDL has a potential regulatory effect on product distribution and enhance the research significance of this work.

4. A key issue involved in CO₂RR under acidic conditions is whether the proton source during proton-coupled electron transfer comes from H⁺ in solution or from H* generated by water decomposition. There have been some recent studies on acidic CO₂RR that suggest that the protons come from the dissociation of water, because a 'higher local pH' will then be present near the catalyst layer during reaction, so from this point of view, the acid CO₂RR is not pseudo-acidic? The statement mentioned in this work that the abundant OH on the catalyst surface plays an important role for CO₂-to-C₂₊ conversion in acid medium is not accurate.

5. Acidic CO₂RR can avoid the production of carbonates, thereby improving the utilization rate of carbon, which is an important index to evaluate the acidic CO₂RR performance of catalysts. The single-pass conversion efficiency (SPCE) of 42% in this work is at a level of what, and the authors should also take this as an indicator in the radar chart.

Reviewer #2:

Remarks to the Author:

The authors report a high faradaic efficiency of 87% for CO₂-to-C₂₊ under a current density of 1.6 A/cm² in acidic electrolyte. The partial current density of multicarbon products reaches -0.34 A/cm^2 under dilute CO₂. These findings are claimed to benefit from the construction of ultrathin superhydrophobic macropore Cu gas diffusion electrode. Multi-physics modelling is used to investigate the diffusion of CO₂ in the GDL and the experiments are carefully designed. The manuscript is generally written well, with most data presented clearly. However, some of the contents require careful revision and clarification before it can be published in Nature Communications:

1. The authors claimed different reaction orders of C₂⁺ formation in acid (pH 1) and neutral (pH 6), with the former being second-order and the latter being first-order reaction. The claim was rationalized by the rate analysis of C₂ formation vs concentration of catalyst and hydroxide. The authors should specify the method for the analysis and include more discussion, as this is one of the major claims in the manuscript. Accurately determining the number of active sites is difficult. Quantitative measurement of ECSA through capacitance relies on the charge density, which depends on the crystal structure and composition of the electrolyte.
2. The evaluation of surface pH requires a second method to verify. The onset potential depends on several factors, especially when the electrodes possess different surface geometry. Mass transport, for example, thin-layer diffusion vs planar diffusion, changes the onset potential without any changes in the surface pH.
3. The diffusion of CO₂ is studied by comparing the model between MCFP vs NCBL. My concern is the link of the physical model to the design of Cu-GDE. Both model consider two-layer MCFP and NCBL, while Cu-GDE only has one layer. The authors need to explain more clearly the link between the modelling results and the rational design of Cu-GDE. Fig 1a is a bit confusing and the author may want to improve it.
4. The idea is to design super hydrophilic electrode, which is successfully demonstrated (Fig 2). For CO₂RR, water is critical as it is the proton source for hydrogenation. A balance of water and CO₂ is the key to high selectivity and activity to C₂⁺. Assessing water to the surface seems very difficult in the current design. Have the authors considered tuning the surface hydrophobicity for CO₂-to-C₂⁺? In Fig. S34, different trend of the FEC₂⁺ for CO₂-50% and CO₂-100% was observed for pH 6 and pH 1, which may indicate mass transport is not the only influence. The authors may want to elaborate more here.

Some minor suggestions:

1. More experimental details should be included for the community to follow the work. For example, as I mentioned before, the rate analysis of C₂⁺ formation. The method employed for the multi-physics modelling is missing. Which software/platform did the author use to obtain the results? In Fig 4h, the authors showed the figure of different sources of carbon — CO₂, CO and carbonate. However, I didn't find the relevant description of differentiating the carbon source in the method section.
2. Some of the figures may need to be revised for better readability. Fig 1a is not straightforward and difficult to read. The symbols in Fig 4 d-f for pH 6 and pH1_CO₂ are too similar to differentiate.

Reviewer #3:

Remarks to the Author:

General Comments:

The authors present an impressive demonstration of an optimized CO₂ reduction device containing a tailor-made porous Cu electrode with optimal transport properties and local pH for active and selective reduction of CO₂ to C₂⁺ products in an acidic electrolyte. The results presented are certainly state-of-the-art, and the reviewers appreciate the authors' rational design approach to developing their porous electrode. Nonetheless, the authors present some significant claims regarding the importance of the porous electrode structure and its roughness. Backing up these claims with operando measurement and observation of the porous electrode during CO₂ electrolysis would be useful to address concerns with and cement the hypotheses presented by the work. Hence, given the excellent performance and well-structured narrative, the reviewer suggests acceptance after the following revisions are addressed.

Specific Revisions:

1. Page 1, Line 8: Multiple articles have demonstrated, to the contrary, that CO₂R occurs at liquid solid boundaries. (<https://pubs.acs.org/doi/abs/10.1021/acscatal.0c03319>,

<https://pubs.acs.org/doi/abs/10.1021/acscenergylett.1c01513>). The reviewer suggests caution when invoking the triple-phase boundary theory. This is especially in these systems and how all ions and reactants may not reach the site.

2. Page 1, Line 2: How does the presence of OH⁻ affect CO₂ adsorption and diffusion? This is not immediately clear. Most reports of CO₂R just suggest that high pH is necessary to suppress HER.

3. Fig 1a: It is odd how the alkaline CO₂RR is shown as consuming OH⁻ when these reactions are net-generating for OH⁻.

4. Page 4, Line 15: The justification of the needle structure via invoking of Laplace pressure is nice, but how do the authors know that the needle-like nanostructures persist under reducing conditions? Also the limitations through the GDL are often not as large as possible very local transport and was this explored? Cu has been known to reconstruct under reducing conditions, so have the authors performed operando imaging to observe the morphology during CO₂R to ensure the authors claims are sound?

5. Page 5, Line 32: Recent modeling has shown that bubbles might not limiting in terms of gas transport, especially within the catalyst layers. Have the authors actually modeled the this to show that bubbles are limiting in terms of CO₂ transport?

6. Page 7, Line 25: How does the acid not neutralize the OH concentration on the Cu-OH catalyst surface? Acid-base recombination should be an incredibly favorable and kinetically facile reaction.

7. Figure 4j: This figure is quite challenging to read. The result would be clearer if one or two metrics were shown as a scatterplot instead and the rest were placed in the SI. As it is currently in the MS, it is very challenging to compare the authors' work to the literature as intended.

8. Page 7, Rate Order Analysis: How can the authors be sure that local CO₂ concentration or pH is not also changing during these analyses with respect to roughness? The change in the local environment could also be contributing to these observed trends rather than the surface concentrations of sites or adsorbed OH. Deconvoluting these trends is needed to have a physically meaningful rate expression.

We are grateful to all the reviewers for their valuable comments and suggestions, which help us to significantly improve the revised manuscript. We have carefully considered all the comments and revised our manuscript accordingly. Below is a point-by-point response to the reviewers' comments.

Point-by-point response to the reviewers' comments

Reviewer #1 (Remarks to the Author):

In this manuscript, the authors first validated CO₂ diffusion efficiency (CO₂DE) of the conventional GDL and found that CO₂DE can be improved by optimizing thickness, pore size and hydrophobicity of GDL. Besides, they further systematically evaluated the CO₂RR performance of the optimal GDL in an acidic environment, which exhibits a Faradaic efficiency of 87% with a partial current density ($j_{C_{2+}}$) of -1.6 A cm^{-2} for multicarbon products (C₂₊). Despite having appreciable and interesting results, there are still some issues that would require a thoughtful and major revision of the manuscript:

Response:

We appreciate the reviewer's comments and suggestions on our work. We have revised the manuscript based on the reviewer's suggestions.

1. It is mentioned in the manuscript that Cu-GDL was obtained by a mild temperature program (3 °C per minute) by gradually heating the porous Cu(OH)₂ to 180 °C over the course of 1 h in an H₂ environment. And then porous Cu was immersed in an Ar-saturated ethanol solution containing 1-octadecanethiol (10 mM, Wako) at 60 °C for 10 min. The results of Raman spectroscopy and XPS show that the surface of Cu-GDL is copper, and there is still a lot of -OH on the surface. Among them, hydrogen heat treatment can reduce Cu(OH)₂ to metallic copper, but how OH is produced and stable. Please provide an explanation in this regard.

Response:

We thank the reviewer for the thoughtful comment. Based on the characterization by XPS and XRD, Cu-GDL was not found to have Cu(OH)₂ composition. However, Cu-GDL exhibited Raman bands similar to those observed for Cu(OH)₂, indicating the presence of OH in Cu-GDL, although not in the form of Cu(OH)₂. According to the XPS results for Cu2p, the Cu2p_{2/3} peak in Cu-GDL was observed at 932.8 eV (Supplementary Figure 12), slightly higher than metallic Cu (932.6 eV) and Cu₂O (932.5 eV) and significantly lower than Cu(OH)₂ (934.3 eV) (10.1021/acscatal.2c03650). Therefore, it is possible that OH is present in the form of Cu(OH)_x, where the oxidation state of Cu is close to 0.

Supplementary Figure 12. XPS spectrum of Cu 2p for porous Cu and Cu-GDL.

2. It is worth mentioning that previous studies have found that alkali metal cations (Na^+ , K^+ , Cs^+) play an important role in inhibiting hydrogen evolution reaction (HER) and improving the selectivity of CO_2RR to C_{2+} products in acidic electrolytes (Science 2021, 372, 1074; Nat. Commun. 2022, 13, 7596; Nat. Catal. 2022, 5, 268). However, the influence of alkali metal cations (K^+) on the HER and CO_2RR is not mentioned in this manuscript. Please specify whether alkali metal cations contribute to CO_2RR of performance in acidic environments in this work?

Response:

As you suggest, our experimental results and previous discussions in literatures indicate that decreasing alkali metal ion concentration does indeed have a direct effect on product selectivity and reaction rates. In recent years, the focus on alkali metal concentration in acidic systems has been primarily on the ease of carbonate formation (10.1126/science.abg6582). The formation of low concentration carbonates helps to improve carbon utilization in CO_2RR , but it is challenging to balance reaction rates and selectivity (10.1038/s41467-022-35415-x; 10.1021/acscenergylett.2c02292). Therefore, we selected an appropriate K^+ concentration for discussion, with the aim of proposing strategies to improve carbon utilization in acidic environments without compromising selectivity and reaction rates.

3. A more detailed distribution of C_{2+} products should be provided in Fig. 4a, which is conducive to further study of whether the optimized Cu-GDL has a potential regulatory effect on product distribution and enhance the research significance of this work.

Response:

We thank the reviewer for this suggestion. We considered showing the specific product distribution in Fig. 4a. However, due to the amount of data, it was difficult to present it clearly within a single figure. Therefore, in Fig. 4a, we have provided a more straightforward representation, while the detailed product distribution has been included in the Supplementary Figure 21 for reference. In addition, the C_{2+} selectivity exhibited by this catalyst is similar to that of most Cu-based catalysts with high C_{2+} selectivity, such as for ethylene and ethanol, which account for approximately 80% of the total C_{2+} products.

Supplementary Figure 21. CO_2RR performance of Cu-GDL with different pH electrolyte. a, pH 6, b, pH 1.

4. A key issue involved in CO₂RR under acidic conditions is whether the proton source during proton-coupled electron transfer comes from H⁺ in solution or from H* generated by water decomposition. There have been some recent studies on acidic CO₂RR that suggest that the protons come from the dissociation of water, because a ‘higher local pH’ will then be present near the catalyst layer during reaction, so from this point of view, the acid CO₂RR is not pseudo-acidic? The statement mentioned in this work that the abundant OH on the catalyst surface plays an important role for CO₂-to-C₂₊ conversion in acid medium is not accurate.

Response:

We thank the reviewer for the thoughtful comment. The points are similar to the discussion of whether the anodic oxygen evolution reaction in CO₂RR originates from the oxidation of OH⁻ or H₂O. Even though we used a high concentration of strong alkaline anolyte, H₂O, as a solvent makes up the majority of the total electrolyte solution. Similarly, in acidic electrolytes, we are faced with the fact that H₂O predominates in the electrolyte solution. Therefore, it is challenging to determine whether the protons come primarily from the acid or from the H₂O. The OH on Cu-GDL can increase the pH of the electrode surface, which suppresses HER and promotes CO₂RR. Therefore, we proposed that "the abundant OH on the catalyst surface plays an important role in CO₂-to-C₂₊ conversion in acidic medium." The investigation of whether the local electrolyte is truly acidic or pseudo-acidic is particularly intriguing and warrants further investigation. It may be influenced by various factors such as the type, pH, and concentration of the electrolyte. In the future, with the assistance of in-situ characterization techniques, we may be able to observe some interesting phenomena.

5. Acidic CO₂RR can avoid the production of carbonates, thereby improving the utilization rate of carbon, which is an important index to evaluate the acidic CO₂RR performance of catalysts. The single-pass conversion efficiency (SPCE) of 42% in this work is at a level of what, and the authors should also take this as an indicator in the radar chart.

Response:

We appreciate the valuable suggestions. We have then added the SPCE indicator to the radar chart and compared its performance with the CO₂RR in an acidic environment over the past two years. These comparisons were made with the data from Science in 2021, Nature Catalysis in 2022, and Nature Synthesis in 2023.

Fig 4j Comparison of FE_{C₂₊}, FE_{H₂}, j_{C₂₊}, SPCE and pH of bulk electrolyte on Cu-GDL with those on the state-of-the-art CO₂RR in acidic media.

Reviewer #2 (Remarks to the Author):

The authors report a high faradaic efficiency of 87% for CO₂-to-C₂₊ under a current density of 1.6 A/cm² in acidic electrolyte. The partial current density of multicarbon products reaches -0.34 A/cm² under dilute CO₂. These findings are claimed to benefit from the construction of ultrathin superhydrophobic macropore Cu gas diffusion electrode. Multi-physics modelling is used to investigate the diffusion of CO₂ in the GDL and the experiments are carefully designed. The manuscript is generally written well, with most data presented clearly. However, some of the contents require careful revision and clarification before it can be published in Nature Communications:

Response:

We thank the reviewer for his/her encouragement of our work. We have revised the manuscript based on the reviewer's comments.

1. The authors claimed different reaction orders of C₂₊ formation in acid (pH 1) and neutral (pH 6), with the former being second-order and the latter being first-order reaction. The claim was rationalized by the rate analysis of C₂ formation vs concentration of catalyst and hydroxide. The authors should specify the method for the analysis and include more discussion, as this is one of the major claims in the manuscript. Accurately determining the number of active sites is difficult. Quantitative measurement of ECSA through capacitance relies on the charge density, which depends on the crystal structure and composition of the electrolyte.

Response:

We thank the reviewer for this important suggestion. We have performed a more detailed analysis and discussion of the relationship between the FR_{C₂₊}, the catalyst, and its OH concentration. We have presented this analysis sequentially under the corresponding figures (Supplementary Figure 24-28) for the convenience of readers.

In addition, you are correct in pointing out that using ECSA to accurately quantify the number of active sites is indeed challenging because it is influenced by multiple factors. However, comparing relationships obtained for the same catalyst in the same test environment remains valuable in assessing active site performance.

Supplementary Figure 24. CO₂RR performance of Cu-GDL for the formation of C₂₊ compounds as a function of different synthesis time. a, C₂₊ partial current density ($j_{C_{2+}}$) and b, C₂₊ formation rate (FR_{C₂₊}).

The calculation of $j_{C_{2+}}$ and FR_{C₂₊} can be referred to Equation 12 and Equation 13.

Supplementary Figure 25. Electrochemical surface areas (ECSA) and roughness factors (rf) for porous Cu with different synthesis time. CV of a, 0 min, b, 10 min, c, 20 min, d, 30 min, e, ECSA and f, rf.

The ratio of the slope/0.445 gave the rf of the surface and the rf value for a Cu foil surface is defined as 1.

Supplementary Figure 26. C_{2+} formation rate as a function of roughness factor in pH 6 and pH 1 electrolyte.

Based on the relationship between $FR_{C_{2+}}$ and rf (Supplementary Figure 26 and Figure 4b), it could be observed that there was a linear relationship between $FR_{C_{2+}}$ and rf at pH 6, while $FR_{C_{2+}}$ showed linear response to rf^2 . The corresponding slopes were calculated from these linear responses.

Supplementary Figure 27. Composition analysis of the electrode. a, Concentration of catalytic sites ([Cat.]) and **b,** OH ([OH]) for each roughness factor.

The concentration of catalytic sites [Cat.] corresponding to the rf of the electrode was calculated by substituting the slope obtained at pH 6 (Supplementary Figure 26) into Equation 8, as shown in Figure 27a. Similarly, the square of the catalyst concentration $[Cat.]^2$ corresponding to the rf of the electrode was obtained by using the slope value obtained at pH 1 (Figure 4b) in Equation 8. Considering that the OH concentration [OH] is related to [Cat.] since the OH originates from the catalyst, [OH] was calculated by dividing $[Cat.]^2$ by [Cat.] obtained from Supplementary Figure 27a, as shown in Supplementary Figure 27b.

Supplementary Figure 28. C₂₊ formation rate analysis. C₂₊ formation rate (FR_{C₂₊}) as a function of **a**, the concentration of catalytic sites ([Cat.]) and **b**, the multiplication of [Cat.] and OH concentration ([Cat.][OH]).

Based on the data obtained from Supplementary Figure 24 for FR_{C₂₊} and from Supplementary Figure 27 for [Cat.] and [OH], these values were inserted into Equations 8 and 9 to calculate the reaction rate constants corresponding to the respective pH. As a result, the reaction rate constant towards Cu-GDL is 4.20 × 10² h⁻¹ at pH 6 (first-order reaction), and 2.45 × 10¹ μmol⁻¹ cm² h⁻¹ at pH 1 (second-order reaction).

2. The evaluation of surface pH requires a second method to verify. The onset potential depends on several factors, especially when the electrodes possess different surface geometry. Mass transport, for example, thin-layer diffusion vs planar diffusion, changes the onset potential without any changes in the surface pH.

Response:

In fact, the use of the onset potential to evaluate the pH of the electrode surface is a method that is used after weighing various influencing factors. We mention "**Given various influential factors such as current density, diffusion layer thickness, bulk electrolyte composition, and other factors, the determination of a pH just above the electrode surface, "intrinsic pH", during CO₂RR appears to be challenging²⁶. To address this, we applied an electrochemical approach based on the onset potential for the oxidation of Cu⁰ to Cu¹, and the intrinsic pH on Cu-GDL was calculated from the onset potential observed on a pristine Cu foil as reference using Nernst equation (see Methods and Supplementary Fig. 20 for details).**" in the manuscript. Considering that the differences in mass transport generated by electrode surfaces with different roughness, we deliberately performed tests at low *j* in the range of μA cm⁻², employing a slow scan rate of 10 mV s⁻¹ to minimize the impact of mass transport on the experimental results as much as possible.

Furthermore, we used a variety of in situ and ex situ characterization techniques, such as Raman and XPS, to confirm the abundant presence of OH on the electrode surface, providing multiple lines of evidence to ensure the high pH of the electrode surface.

3. The diffusion of CO₂ is studied by comparing the model between MCFP vs NCBL. My concern is the link of the physical model to the design of Cu-GDE. Both model consider two-layer MCFP and NCBL, while Cu-GDE only has one layer. The authors need to explain more clearly the link between the modelling results and the rational design of Cu-GDE. Fig 1a is a bit confusing and the

author may want to improve it.

Response:

We completely understand of the reviewer's concerns. Our modeling actually considers three layers: MCFP, NCBL, and the catalyst layer, and we've analyzed the impact of each layer on CO₂ diffusion. We proposed three ways to enhance CO₂ diffusion, including reducing the thickness of the GDL, enlarging the pores of GDL, and designing a superhydrophobic catalyst structure. Based on this, we introduced Cu-GDL, which effectively integrates these three layers into one, thus eliminating the limitations on CO₂ diffusion that are present in conventional gas diffusion electrodes. In response to the reviewer's suggestion, we have added transition sentences in the manuscript to help the readers' understanding: "This assessment of GDL architectures led us to design a GDE with a thinner, macroporous diameter, as well as vertical and needle-like surface structures to facilitate CO₂ diffusion." (page 3)

Furthermore, as suggested by the reviewer, we acknowledge that the description of CO₂ diffusion limitations in Fig 1a was not direct enough. We have reconstructed Fig. 1a to better illustrate the factors influencing CO₂ diffusion.

Fig. 1 a Illustrations of CO₂ diffusion in the most common GDL. The thickness and pore diameter of the GDL, and the hydrophobicity of the catalyst together affect the CO₂ diffusion. (MCFP is macroporous carbon fiber paper and NCBL is nano-microporous carbon black layer).

4. The idea is to design super hydrophilic electrode, which is successfully demonstrated (Fig 2). For CO₂RR, water is critical as it is the proton source for hydrogenation. A balance of water and CO₂ is the key to high selectivity and activity to C₂₊. Assessing water to the surface seems very difficult in the current design. Have the authors considered tuning the surface hydrophobicity for CO₂-to-C₂₊? In Fig. S34, different trend of the FEC₂₊ for CO₂-50% and CO₂-100% was observed for pH 6 and pH 1, which may indicate mass transport is not the only influence. The authors may want to elaborate more here.

Response:

We appreciate the reviewer's useful comments, which have been discussed in the literature (doi.org/10.1002/aenm.202103663). The authors of that study found that electrodes with optimal hydrophobicity exhibited the best CO₂RR performance. Similarly, we also compared the effects of 1-hexanethiol, 1-dodecanethiol, and 1-octadecanethiol-modified Cu electrodes on CO₂RR performance. The results indicated that the superhydrophobic Cu electrode modified with 1-octadecanethiol (the electrode described in this manuscript) exhibited the best selectivity and reaction rate. Thus, from these results, it appears that superhydrophobic electrodes are more

conductive to enhancing selectivity compared to balanced hydrophobic and hydrophilic electrodes. Additionally, we suggest that the trends in FEC_{2+} observed at 100% CO_2 and 50% CO_2 under pH 6 and pH 1 conditions are similar. The differences observed at pH 6 and pH 1 can be both attributed to the absence of OH^- in strong acidic electrolytes, affecting CO_2 diffusion rates. However, at 50% CO_2 , the peak j is lower, which is why we observed a decreasing trend in FEC_{2+} at lower j .

Supplementary Figure 34. CO_2RR performance of Cu-GDL with different CO_2 concentrations. a-b, pH 6 and c-d, pH 1. a,c, FEC_{2+} and b,d, $j_{\text{C}_{2+}}$.

Some minor suggestions:

1. More experimental details should be included for the community to follow the work. For example, as I mentioned before, the rate analysis of C_{2+} formation. The method employed for the multi-physics modelling is missing. Which software/platform did the author use to obtain the results? In Fig 4h, the authors showed the figure of different sources of carbon — CO_2 , CO and carbonate. However, I didn't find the relevant description of differentiating the carbon source in the method section.

Response:

Regarding the analysis of the C_{2+} formation rate and the corresponding reaction order with respect as a function of pH, we have addressed this in response to the reviewer's first suggestion.

Response:

In our modeling approach, we considered that the factors influencing CO_2 diffusion vary significantly at different locations, from its initial entry through the gas diffusion electrode to its eventual participation in the CO_2RR . To evaluate the impact of these factors at different locations, we employed Fick's second law, Knudsen diffusivity, and Laplace pressure, resulting in the conclusion that ultra-thin, macroporous, and super-hydrophobic gas diffusion electrode provide the optimal environment for CO_2 diffusion. It's possible that multi-physics modeling may not directly

influence our strategy for designing gas diffusion electrode.

In addition, our modeling approach is relatively straightforward and does not require any specialized software or platforms for computation. It is performed using the same software that is commonly used for the other data analyses, which is why it was not specifically mentioned in the manuscript.

Response:

We have included a description of the carbon source attribution below Supplementary Table 2 to help readers quickly understand this part of the analysis process.

Supplementary Table 2. Theoretical maximum SPCE of CO₂ at a flow rate of 3.5 sccm.

Product	Theoretical maximum SPCE (%)			
	pH 6	pH 6_CO ₂	pH 1	pH 1_CO ₂
Formic acid	5.13	6.96	6.14	5.32
Carbon monoxide	3.72	4.54	5.20	6.53
Methane	0.08	0.17	0.140	0.19
Ethylene	9.18	9.25	9.45	9.08
Ethane	0.002	0.001	0.002	0.002
Ethanol	7.53	6.34	6.14	6.27
Acetic acid	3.56	3.53	3.44	3.21
n-Propanol	1.09	0.80	0.99	1.08
Total	30.29	31.59	31.49	31.68

Equation 17 can be used to calculate the SPCE for each target product in Supplementary Table 2. "Total" in Supplementary Table 2 represents the total SPCE associated with the CO₂ conversion to products. The difference between the experimentally obtained SPCE from Figure 4d and the "Total" in Supplementary Table 2 represents the fraction of CO conversion into products beyond the theoretical maximum. The remaining fraction that has not been converted into products is the fraction contributing to carbonates formation.

2. Some of the figures may need to be revised for better readability. Fig 1a is not straightforward and difficult to read. The symbols in Fig 4 d-f for pH 6 and pH1_CO₂ are too similar to differentiate.

Response:

We appreciate the reviewer’s suggestions. In response to the suggestion, we have improved and made more explicit description of the constraints on CO₂ diffusion in Fig 1a. Additionally, in Figures 4d-f, we have replaced the symbols representing pH1_CO₂ with Δ to better distinguish them from

other symbols.

Fig. 1 a, Illustrations of CO₂ diffusion in the most common GDL. The thickness and pore diameter of the GDL, and the hydrophobicity of the catalyst together affect the CO₂ diffusion. (MCFP is macroporous carbon fiber paper and NCBL is nano-microporous carbon black layer).

Fig 4 d, SPCE, **e**, CO selectivity and **f**, C₂₊ selectivity as a function of CO₂ flow rate at an applied j of -0.5 A cm^{-2} using electrolytes with pH=6 (pH 6) and pH=1 (pH 1) and CO₂ saturated electrolytes with pH=3.8 (pH 6_CO₂) and pH=0.9 (pH 1_CO₂).

Reviewer #3 (Remarks to the Author):

General Comments:

The authors present an impressive demonstration of an optimized CO₂ reduction device containing a tailormade porous Cu electrode with optimal transport properties and local pH for active and selective reduction of CO₂ to C₂₊ products in an acidic electrolyte. The results presented are certainly state-of-the-art, and the reviewers appreciate the authors' rational design approach to developing their porous electrode. Nonetheless, the authors present some significant claims regarding the importance of the porous electrode structure and its roughness. Backing up these claims with operando measurement and observation of the porous electrode during CO₂ electrolysis would be useful to address concerns with and cement the hypotheses presented by the work. Hence, given the excellent performance and well-structured narrative, the reviewer suggests acceptance after the following revisions are addressed.

Response:

We are grateful to the reviewer for his/her positive evaluation of our work. We have revised our paper based on the reviewer's suggestions.

Specific Revisions:

1. Page 1, Line 8: Multiple articles have demonstrated, to the contrary, that CO₂R occurs at liquid solid boundaries. (<https://pubs.acs.org/doi/abs/10.1021/acscatal.0c03319>, <https://pubs.acs.org/doi/abs/10.1021/acsenerylett.1c01513>). The reviewer suggests caution when invoking the triple-phase boundary theory. This is especially in these systems and how all ions and reactants may not reach the site.

Response:

We thank the reviewer for the comment. We have removed the description of the triple-phase boundary in the manuscript. As the reviewer correctly pointed out, the catalyst applied to the gas diffusion electrode surface can be either hydrophilic or poorly hydrophobic, so the use of the triple-phase boundary description may not be accurate in this context.

2. Page 1, Line 2: How does the presence of OH⁻ affect CO₂ adsorption and diffusion? This is not immediately clear. Most reports of CO₂R just suggest that high pH is necessary to suppress HER.

Response:

We thank the reviewer for the comment. CO₂ is an acidic gas molecule, and in alkaline electrolytes, OH⁻ acts as a facilitator for CO₂ diffusion. Conversely, the more acidic the system, the more CO₂ diffusion is restricted. This is reflected in the phenomenon that the partial current density of C₂₊ at pH 1 is lower than that at pH 6, although the selectivity for C₂₊ and HER is similar. The same phenomenon has also been observed in previously published literature. For instance, similar electrodes from the same research group exhibited significantly lower current density in acidic environments compared to alkaline environments (10.1126/science.abg6582, 10.1126/science.aay4217). Therefore, we conclude that OH⁻ in the electrolyte promotes the CO₂ diffusion.

3. Fig 1a: It is odd how the alkaline CO₂RR is shown as consuming OH⁻ when these reactions are net-generating for OH⁻.

Response:

We apologize for any confusion this may have caused the reviewer. Here, we would like to describe the production and consumption of OH⁻ using the example of CO₂RR to produce CO.

a. Production of OH⁻ in CO₂RR

CO₂RR generates OH⁻ as follows: $\text{CO}_2 + \text{H}_2\text{O} + 2\text{e}^- \rightarrow \text{CO} + 2\text{OH}^-$ (1)

b. Consumption of OH⁻ in CO₂RR

Consumption of OH⁻ produced in (1) to form CO₃²⁻: $\text{CO}_2 + 2\text{OH}^- \rightarrow \text{CO}_3^{2-} + \text{H}_2\text{O}$ (2)

The net result of reactions (1) and (2) is that the rate of OH⁻ production is zero.

c. Consumption of OH⁻ through absorption reactions

OH⁻ can react with CO₂ through absorption reactions in an alkaline electrolyte, which consumes OH⁻: $\text{CO}_2 + 2\text{OH}^- \rightarrow \text{CO}_3^{2-} + \text{H}_2\text{O}$ (3)

In summary, CO₂RR in an alkaline electrolyte results in the consumption of OH⁻.

4. Page 4, Line 15: The justification of the needle structure via invoking of Laplace pressure is nice, but how do the authors know that the needle-like nanostructures persist under reducing conditions? Also the limitations through the GDL are often not as large as possible very local transport and was this explored? Cu has been known to reconstruct under reducing conditions, so have the authors performed operando imaging to observe the morphology during CO₂R to ensure the authors claims are sound?

Response:

To address to this concern, we have performed morphological characterization of the post-reaction electrode, which confirms the stable presence of needle-like nanostructures after CO₂RR (Supplementary Figure 22).

Response:

The reviewer's consideration makes a lot of sense. Compared to CO₂ diffusion through a gas diffusion electrode, the effect of the local environment on CO₂ diffusion is more pronounced as we described in the manuscript, "**Considering that the CO₂ diffusion coefficient in the gas phase is approximately four orders of magnitude higher than that in the liquid phase.**" We have also experimented with Cu electrodes modified with 1-hexanethiol, 1-dodecanethiol, and 1-octadecanethiol, which produce different diffusion layer thicknesses on the catalyst surface. The results indicated that the Cu electrode modified with 1-octadecanethiol (which is the electrode proposed in this manuscript) exhibits the highest reaction rate.

Response:

The reviewer's suggestion is quite reasonable. Indeed, the possibility of Cu restructuring during the CO₂RR process has indeed been reported in the literature. Therefore, we have conducted a morphological characterization of the electrode after the reaction, and SEM analysis confirmed that the needle-like nanostructures did not undergo any restructuring during the reaction. If a restructuring reaction occurs, the hydrophobic layer on the Cu-GDL surface may be immediately disrupted, making efficient CO₂RR unlikely. Therefore, we believe that Cu-GDL used in this study did not undergo restructuring during the CO₂RR process.

Supplementary Figure 22. Morphological characterization of Cu-GDL after operating CO₂RR at j from 0.3 to 1.8 A cm⁻². A pH 6, B pH 1.

Cu-GDL after operating CO₂RR maintains the original morphology.

5. Page 5, Line 32: Recent modeling has shown that bubbles might not limiting in terms of gas transport, especially within the catalyst layers. Have the authors actually modeled the this to show that bubbles are limiting in terms of CO₂ transport?

Response:

We thank the reviewer for the thoughtful consideration. We have removed the part regarding the impact of bubbles on CO₂ diffusion. We suspect that bubbles may not necessarily restrict CO₂ diffusion on the electrode surface under the strong CO₂ flow condition here in this study and further research is needed to better understand the influence of bubbles on diffusion in the future.

6. Page 7, Line 25: How does the acid not neutralize the OH concentration on the Cu-OH catalyst surface? Acid-base recombination should be an incredibly favorable and kinetically facile reaction.

Response:

Although the acid-base recombination should be a highly favorable and kinetically facile reaction, Cu-GDL maintains high efficiency and selectivity even during CO₂RR for a long time in highly acidic environments. This suggests that acid-base recombination may not take place at the electrode surface. We propose that the reason for this could be the presence of a hydrophobic alkanethiol layer on our catalyst surface, which acts as a barrier and limits the occurrence of acid-base recombination reactions.

7. Figure 4j: This figure is quite challenging to read. The result would be clearer if one or two metrics were shown as a scatterplot instead and the rest were placed in the SI. As it is currently in the MS, it is very challenging to compare the authors' work to the literature as intended.

Response:

In response to the reviewer's advice that the radar chart contained an excessive amount of comparative information, we have simplified it and compared our data with the representative literature on CO₂RR in acidic electrolyte environments from the past two years, specifically the articles in Science from 2021, Nature Catalysis from 2022, and Nature Synthesis from 2023.

Fig 4j, Comparison of $FE_{C_{2+}}$, FE_{H_2} , $j_{C_{2+}}$, SPCE and pH of bulk electrolyte on **Cu-GDL** with those on the state-of-the-art CO_2RR in acidic media.

8. Page 7, Rate Order Analysis: How can the authors be sure that local CO_2 concentration or pH is not also changing during these analyses with respect to roughness? The change in the local environment could also be contributing to these observed trends rather than the surface concentrations of sites or adsorbed OH. Deconvoluting these trends is needed to have a physically meaningful rate expression.

Response:

We thank the reviewer for the insightful comment. We would like to discuss these factors in two different pH environments, pH 6 and pH 1.

At pH 6, increasing electrode roughness could potentially enhance CO_2 diffusion, leading to an increase in local CO_2 concentration. This is because the hydrophobic needle-like Cu catalysts act as carriers for CO_2 diffusion. Meanwhile, it's important to note that the catalyst also acts as a catalyst for the reaction itself, so any increase in CO_2 concentration may be offset by the consumption of CO_2 in the catalytic reaction. As a result, the observed results are similar to what would be expected if only the effect of catalyst amount on reaction rate were considered.

At pH 1, increasing electrode roughness not only increases the catalyst concentration, as mentioned earlier, but also leads to an increase in OH concentration. This increase in the OH concentration effectively corresponds to an increase in local pH. We have considered this aspect and found it to be a significant factor influencing the rate of CO_2RR at pH 1.

In summary, we have proposed that both the catalyst and its OH concentration contribute to the reaction kinetics.

Reviewers' Comments:

Reviewer #1:

Remarks to the Author:

All concerns are solved, and I recommend it to be published.

Reviewer #2:

Remarks to the Author:

All my comments are well addressed.

Reviewer #3:

Remarks to the Author:

The Authors have adequately addressed the comments raised although several of the responses noted that they have said it is outside the scope or for future study and have removed text and I would be interested in them adding a short summary paragraph aggregating the issues and ideas that need further study at the end of the manuscript.

Reviewer #1 (Remarks to the Author):

All concerns are solved, and I recommend it to be published.

Response:

We thank the reviewer for your prompt response and assistance. We are pleased to hear that all concerns have been addressed, and appreciate your recommendation for publication.

Reviewer #2 (Remarks to the Author):

All my comments are well addressed.

Response:

We are pleased to hear that all comments have been well addressed. We thank the reviewer for his/her kind attention.

Reviewer #3 (Remarks to the Author):

The Authors have adequately addressed the comments raised although several of the responses noted that they have said it is outside the scope or for future study and have removed text and I would be interested in them adding a short summary paragraph aggregating the issues and ideas that need further study at the end of the manuscript.

Response:

We appreciate the reviewer's suggestions. We have included a brief summary expressing our thoughts on further research and potential characterization methods. **“In the future, it would be useful to further investigate the influence of the local environment in the gas diffusion electrode using advanced in-situ detection methods.”**